# Using Low-Resolution Non-Invasive Infrared Sensors to Classify Activities and Falls in Older Adults

**DOI:** 10.3390/s22062321

**Published:** 2022-03-17

**Authors:** Gastón Márquez, Alejandro Veloz, Jean-Gabriel Minonzio, Claudio Reyes, Esteban Calvo, Carla Taramasco

**Affiliations:** 1Departamento de Electrónica e Informática, Universidad Técnica Federico Santa María, Concepción 4030000, Chile; 2Escuela de Ingeniería Civil Biomédica & Centro de Investigación y Desarrollo en Ingeniería en Salud, Universidad de Valparaíso, Valparaíso 2340000, Chile; alejandro.veloz@uv.cl; 3Escuela de Ingeniería Informática & Centro de Investigación y Desarrollo en Ingeniería en Salud, Universidad de Valparaíso, Valparaíso 2340000, Chile; jean-gabriel.minonzio@uv.cl; 4Ecoframe SpA, Temuco 4780000, Chile; claudio.reyes@ecoframe.cl; 5Society and Health Research Center, Laboratory on Aging and Social Epidemiology & Millennium Nucleus on SocioMedicine, Facultad de Ciencias Sociales y Artes, Universidad Mayor, Santiago 7560908, Chile; esteban.calvo@umayor.cl or; 6Department of Epidemiology, Mailman School of Public Health, Columbia University, New York, NY 10032, USA; 7Robert N. Butler Columbia Aging Center, Columbia University, New York, NY 10032, USA; 8Escuela de Ingeniería Informática, Universidad de Valparaíso & Millennium Nucleus on SocioMedicine, Valparaíso 2340000, Chile

**Keywords:** fall, older adult, infrared sensor

## Abstract

The population is aging worldwide, creating new challenges to the quality of life of older adults and their families. Falls are an increasing, but not inevitable, threat to older adults. Information technologies provide several solutions to address falls, but smart homes and the most available solutions require expensive and invasive infrastructures. In this study, we propose a novel approach to classify and detect falls of older adults in their homes through low-resolution infrared sensors that are affordable, non-intrusive, do not disturb privacy, and are more acceptable to older adults. Using data collected between 2019 and 2020 with the eHomeseniors platform, we determine activity scores of older adults moving across two rooms in a house and represent an older adult fall through skeletonization. We find that our twofold approach effectively detects activity patterns and precisely identifies falls. Our study provides insights to physicians about the daily activities of their older adults and could potentially help them make decisions in case of abnormal behavior.

## 1. Introduction

According to studies conducted at the World Health Organization [1], a large proportion of the world’s population will be able to aspire to live beyond the age of 60 due to a marked reduction in mortality in the early stages of life, especially during infancy and childbirth, and in mortality from infectious diseases. In this context, people aged 65 and over account for 9.10% of the world’s population. This percentage has almost doubled in the last six decades, from less than 5% in 1960. In absolute numbers, this age group has grown from 150 million in 1960 to 697 million in 2019.

With advancing age, diseases begin to appear, characterized by a gradual deterioration of older adults’ physical and mental health [2], where falls are the leading cause of injury and death among older adults. It is estimated that half of older adults who fall indoors are unable to get up again, with consequences such as pressure ulcers, loss of muscle mass, dehydration, pneumonia, and death [3]. Furthermore, on a psychological level, many older adults develop a fear of falling after a fall, which further limits their daily activities.

Advances in technology and lower hardware costs have made it possible to address the problems of older adults through information technologies from a preventive, rather than reactive, perspective [4]. In this scenario, the concept of smart homes emerges, which are comfortable environments that possess environmental intelligence and automatic control in a home allowing responses to be given to the behavior of the residents [5]. The main functions of smart homes are as follows: serving users with physical disabilities; supporting people with visual impairment and/or deafness; monitoring physiological signs and therapeutic devices; applying assisted therapy; intelligent management of devices, among others.

The technologies that support smart homes range from electronic medical devices to sensors. In this regard, presence sensors are electronic devices that run internal systems that detect movement in the area or environment where they are located [6]. In smart homes, these sensors are often used to optimize the energy consumption and efficiency of diverse systems such as ventilation, lighting, or air conditioning in the home. On the other hand, they are also used for health purposes. Some sensors, such as accelerometers, barometers, inertial sensors, and gyroscopes, provide essential data and information for the care of people, especially older adults. Concerning falls, several studies, such as [7,8,9], agree on the benefits of using sensors to detect falls. Nevertheless, they also discuss the challenges of using sensors in homes inhabited by older adults. Some of these challenges point to the limited ability of some sensors to detect falls in volatile environments (e.g., darkness), high dependence on video cameras to detect falls, and invasion of privacy in the older adult’s home.

Because of this scenario, we proposed eHomeseniors [10], a platform that allows continuous analysis of the daily activities of older adults to detect risk events within the home, such as (i) abnormal behavioral patterns associated with possible neuro- degenerative problems such as repetitive actions and wandering, (ii) falls that significantly degrade the quality of life of the elderly and their family, and (iii) harmful carbon dioxide levels. Our practical experience in using the platform with patients has shown promising results in detecting falls and improving the quality of life of older adults. Nevertheless, we have realized that the technical infrastructure and sensors used to improve fall detection are intrusive to the privacy of older adults. In some cases, this situation has led to the older adult’s family members not approving of the use of technology within their homes.

In this paper, we describe an extension of the eHomeseniors platform which addresses human behavior by capturing, analyzing, and permanently monitoring basic and instrumental activities of daily living in older adults through non-intrusive and low-resolution thermal infrared sensors. These sensors are specially designed for object detection, classification, and positioning, and the detection of shapes, colors, and surface differences, even under extreme environmental conditions. We analyzed two methods to evaluate the daily activity of the older adult, which are the analysis of the presence of the older adult in rooms and skeletonization techniques. We applied our proposal to older adults who are users of the eHomeseniors platform and analyzed the results obtained.

The paper is structured as follows: Section 2 describes related work; Section 3 details the eHomeseniors platform; Section 4 describes our study design; Section 5 illustrates the results; Section 6 describes the discussion and findings; Section 7 concludes and describes future work.

## 2. Related Work

Using sensors to detect, react to, and mitigate falls is relevant to improving the quality of life of older adults [11,12,13,14,15]. In this study, we take a novel approach to classify and detect falls of older adults in their homes through low-resolution infrared sensors that are affordable, non-intrusive, do not disturb privacy, and are more acceptable to older adults. We fill a gap in previous work, which has been largely limited to discussing whether sensors disturb or inconvenience older adults’ privacy, proposing solutions that require an extensive infrastructure to operate and ultimately are not widely accepted by older adults.

De Miguel et al. [11] proposed the use of a fall detector based on a high-precision machine learning algorithm. Through background subtraction techniques, the authors’ proposal, together with optical flow techniques, empirically demonstrated that they achieved the effectiveness of more than 96% on a sample of 50 videos analyzed when detecting falls.

Nahian et al. [12] discussed different methods and systems designed for detecting falls in older adults based on non-contact and wireless sensors. The authors presented a technical and in-depth look at the different technologies and described the definition of a model that responds to fall monitoring and control needs for adults and/or people with pathology that require specific care.

Galvao et al. [13] described methods of detecting and classifying falls in older adults to improve response times and, in turn, reduce the risk of prolonged unattended injury and death of patients. The authors focused on solving problems such as reducing false positives and the overload that this system defect produces. For this, the use of different topologies of a neural network trained to detect falls using RGB images in conjunction with accelerometers was proposed. Proofs of concept for the proposed model have been performed with the UR-Fall and UP-Fall datasets under a comparative perspective with other models.

Azeem et al. [14] proposed a solution in managing privacy and security in the aggregation of patient data and sensor-generated data in remote, efficient, and real-time patient monitoring and healthcare technology-based environments. The authors presented a novel efficient and secure data transmission and aggregation (ESDTA) scheme that improves the delivery of healthcare parameters by using the secure message aggregation (SMA) algorithm in combination with a secure message decryption (SMD) algorithm. The proposal aims to both preserve data integrity and protect the system from various security threats. For this purpose, the simulation of environments using NS 2.35 shows that aggregation at the MN effectively reduces transmission and communication costs. Additionally, it minimizes storage and computation costs in the cloud server, which reduces energy consumption costs, improving resilience and data security.

Santos et al. [15] proposed using convolutional neural networks (CNNs) for fall detection in an IoMT environment, where the data obtained by sensors is encrypted and sent via MQTT and AMQP to fog devices (FN). The research focused on using accelerometers in conjunction with deep learning algorithms to implement three layers, two maxpool, and three fully connected layers as a deep learning model. In this study, Matthews’ accuracy, precision, sensitivity, specificity, and correlation coefficients are used to evaluate its performance. Furthermore, its performance is evaluated by conducting two experiments: the first one uses data aggregation (DA) and the second one uses accelerometers in a smartwatch. Additionally, for this research, three open datasets are used to compare results with other research.

## 3. The eHomeseniors Platform

EHomeseniors (http://www.ehomeseniors.cl, accessed on 7 March 2022) is a non-invasive monitoring platform aimed at preventing falls and caring for older adults both at home and outside the home by installing sensors and portable devices explicitly designed for emergencies [10]. The platform has a notification platform that alerts family members or those close to them in real time via their smartphone.

The platform (see Figure 1) is installed in each home of the older adult (A). In turn, the platform consists of the following components: non-invasive, non-intrusive components of nocturia sensors (B), falling sensors (C), humidity and carbon monoxide sensors (D), and an emergency button to call relatives and family (E). Furthermore, the platform is connected with emergency and firefighter departments (F) through another platform that generates early warnings (1), collects medical and environmental data (2), and provides personal care if it is required (3).

The platform uses an ODROID minicomputer to process signals captured by sensors. This kind of minicomputer is capable of executing artificial intelligence algorithms to detect actimetry and monitoring the concentrations of harmful gases, temperature, and humidity every 5 min. Furthermore, it has processes to send alerts to the centralized server, informing about events such as falls or abnormal concentrations of harmful gases. The platform’s stakeholders can access it using web or mobile applications. Both interfaces illustrate data regarding the follow-up of the elderly, alerts issued, and descriptive statistics of the evolution of the elderly.

Since the platform is non-intrusive, sensors should not interfere in the daily life of the patient. Therefore, the platform uses discreet sensors so that older adults can perform their daily activities without feeling monitored. The older adult may forget the feeling that she or he is being monitored, but at the same time, she or he must know that they are being monitored. In this regard, we inform the older adult where the sensors are situated in their home. Table 1 describes, at a high level, the characteristics of the sensors used by the platform.

The most relevant feature of the platform is the ability to have remote preventive actions for the control of patients. These features, such as shutting off gas to cookers, kitchens, closing or opening doors for emergency services, activation of sound alerts, gas control, and use in the kitchen, should be implemented in real time and prioritized. Therefore, the platform needs efficient bandwidth management and data transmission in order to enable scalability and secure and constant data delivery. In addition, the platform also needs to monitor all the places where the patient moves around in order to achieve greater efficiency in the monitoring of environmental variables, accidents, and health states. In this regard, the platform considers an IoT network, based on the MQTT protocol, using SDN network architecture to manage the platform under IoT standards in a way that is efficient, lightweight, and secure. This configuration allows to raise alerts and perform the help and monitoring actions expected by design, with the maximum use of the infrastructure.

## 4. Research Design

### 4.1. Background

We aim to propose non-invasive and non-intensive fall monitoring techniques using low-resolution infrared sensors. Therefore, the research questions of our study are as follows:RQ1: *Can data obtained from infrared temperature sensors be used to classify basic activities*
*of older adults in their homes*?RQ2: *Can infrared images be represented as simple position-size invariant graphs to detect falls of older adults in their homes*?

### 4.2. Design

Since the study subjects were older adults, we aimed to create an easy-to-install infrastructure that did not require home modifications. Participants in the study must satisfy the following inclusion criteria: be over 65 years of age, not have a life expectancy of fewer than 6 months, live alone most of the time, and not live with pets such as dogs or cats. We also used the following exclusion criteria for older adults: with dementia, a history of alcoholism or substance abuse, unable to understand or sign the informed consent, or who do not wish to participate in the project. Additionally, Table 2 describes the details of the older adults participating in our study. We used the Barthel index [16] in order to measure the older adult’s ability to perform ten activities of daily living considered fundamental by obtaining their degree of independence. In addition to the Barthel index, we applied other instruments to classify and select older adults for our study. Details of the instruments used can be found in Appendix B. For the purposes of this study, we spanned data from 2019 to 2020.

Concerning the installation of the sensors, the distribution of the sensors is the same in all the houses; one infrared sensor located in the living/dining room and another sensor located in the older adult’s bedroom. The lower sensor is located 30 cm above the floor and the upper sensor 100 cm above the floor, both positioned in the corner of the room. For example, Figure 2 depicts how we distributed the sensors within the home of an older adult participating in our study.

### 4.3. Activity Detection

Concerning RQ1, we developed an activity measurement algorithm in which the principle is to compare the temperature of each pixel of the sensor to the background temperature used as a reference. If this difference is larger, to a set threshold, an activity is detected [17]. Indeed, this reference temperature changes gradually, inhibiting the setting of a single threshold. However, the characteristic time of evolution of the background temperature, usually of the order of tenths of minutes, is larger in comparison to the characteristic time of older adults move. Therefore, we choose to use the weighted moving average (WMA) [18] as a simplistic and robust low-pass filter in order to obtain the reference, or background, temperature. The detailed description of the infrared sensors that capture thermal images is described in Table 3.

For 1D capture, it is necessary to have four Omron D6T-8L-06 sensors per room in order to capture the entire body of the older adult (see Figure 3). On the other hand, 2D sensors are capable of capturing the movement of an older adult with a single sensor (see Figure 4). The captured data is stored in structured files as 32 ⇥ 24 matrices (2D sensor) and 1 ⇥ 33 arrays (four 1D sensor measurement + timestamp), and then uploaded to a server.

Regarding data selection, it is required that the data capture be executed without interruption for at least 1 h. For this reason, it is possible to have a score value close to reality, i.e., a 1 h capture file needs to have at least 57,600 records for the 2D sensor and 18,000 records for the 1D sensor in order to represent real information.

Each sensor uses a different algorithm because the data capture (frames) is performed differently (1D vs. 2D), so the calculation of the activity score differs for each case. In this regard, Algorithm A1 (see Appendix A) captures data using a pandas (https://pandas.pydata.org (accessed on 7 March 2022))-based dataset. This dataset contains 33 columns corresponding to the timestamp and 32 to measurements. Algorithm A2 (see Appendix A) capture data which is handled directly as 32 ⇥ 24 matrices in numpy (https://numpy.org (accessed on 7 March 2022)). The dataset consists of an array containing the number of frames (matrices) captured in a given h. Finally, Algorithm A3 (see Appendix A) calculates the activity score.

The datasets generated in Algorithm A1 are composed of 33 characters, while Algorithm A2 generates 32 lines of 24 characters. These lines are fixed arrays with several decimal values. However, the last five digits of the arrays generated by the sensors correspond to non-significant zero values for measurement purposes, which increase the volume of data for storage and transport by 20%. Therefore, in order to optimize the system’s performance concerning RQ1, we developed two algorithms to compress (Algorithm A4, more details in Appendix A) and decompress (Algorithm A5, more details in Appendix A) the data used in the activity score calculation. For both arrays (1D and 2D), the outputs are assigned to variable files defined as score dataset with a weight of 763 bytes and filtered_dataset with a weight of 45,902 bytes. Using Algorithm A4, an array of 370 bytes for *score dataset* and 22,200 bytes for filtered_dataset are obtained. We also produce 51.5% as average compression rate for 1D array and 51.64% for 2D with an average CPU usage time of 0.009 s. The compression algorithm is applied after the filtered_dataset array formation process to optimize the process, generating a new array called compressed_dataset, which will be available and consumed by the messaging process. The decompression process is performed in the subscriber; therefore, on some occasions, it will be convenient to decompress data using Algorithm A5.

Additionally, to complement the data capture from the sensors described in Table 3, we use the temperature sensor DHT11, which uses a capacitor humidity sensor and a thermistor to measure the surrounding air and displays the data as a digital signal on the data pin (see Figure 5). This sensor allows us to detect presence when the senior is in the kitchen, identifying daily patterns. The data received from the sensors is stored in a database where the eHomeseniors platform can access it. The stored data contain the date and time the measurement was taken.

### 4.4. Fall Detection

Concerning RQ2, we developed a component for the detection of falls in the homes of older adults. This component transforms a series of input infrared images on a given time span into a simple position–size-invariant graph representation of a person that is moving in front of the 2D infrared sensors. This component determines whether a person is suffering a fall using the well-known Graph2Vec artificial neural network [19], which was specifically developed for embedding graph data into a *k*-dimensional vector space in which simple classification algorithms can be applied to separate graphs categories, i.e., graphs corresponding to a person that is suffering a fall or to a person that is not falling.

The first step in our framework used for fall detection is to represent the infrared 2D measurement as binary images. In our implementation, a user-defined threshold was set experimentally (according to plausible infrared intensities of humans). We empirically found that this approach performs as well as determining this threshold according to the Otsu statistical criterion that is typically used for this task. We preferred the user-defined criterion in order to reduce the computational burden.

We used a series of morphological operations in binary images computed on each time frame with the purpose of preparing data provided by the sensors for the classification into the falling and non-falling classes. Thus, the binary objects within the images were transformed into their single-pixel-wide skeleton representations [20].

Skeletonization is commonly used in morphological image processing for determining rough features of the objects present in the image, e.g., the length of an object, differences in object shapes, among others [21]. Skeletonization can be applied to binary images containing regions of any shape, accomplishing better results for elongated-shaped objects as is the case of a person present in the IR images.

Skeletons are the center line of an object that are obtained as a two-step process by repeated thinning and pruning skeleton candidates that do not contribute to preserve connectivity. These two steps are repeated iteratively until convergence. This work resorts to the widely used algorithm proposed by Zhang and Suen [22,23]. An example result of these steps is presented in Figure 6.

The next step is to represent the obtained skeletons as graphs. In this regard, the obtained skeletons were subsampled in order to define graph nodes. The vertices of the graphs were determined according to connected components in the non-subsampled skeleton. Additionally, the graphs (subsampled skeletons) were connected along the selected time span for combining into a single graph the temporal dynamic of the object and its morphology.

The final step is to evaluate the Graph2Vec neural network [19]. Graph2Vec is a feedforward neural network that is used to compute the embedding of a set of graphs, i.e., the transformation of graphs to a set of vectors that can be classified using simple classification algorithms. The vector embedding is obtained by training Graph2Vec on 120 falls and 120 non-falls acquired from ten subjects. The obtained vectors are finally classified into the falling and non-falling classes using the random forest technique. The number of dimensions *k*, i.e., number of neurons within the hidden layer of Graph2Vec, was set to 30. This value was determined by a grid search strategy. Figure 7 summarizes the steps of the proposed fall detection method.

### 4.5. Plan Validity

We identify potential threats to the internal, external, conclusion, and construct validity of our study. In the following sections, we describe the threats and how we mitigate them.

#### 4.5.1. Construct Validity

The main threat detected is the apprehension of older adults to be evaluated. It is natural for older adults to be afraid of being evaluated with instruments and technological infrastructure within their homes. To mitigate this threat, we have mentioned to patients that the results obtained in this study are strictly for research purposes. Throughout the process of explaining the objectives of our study, a medical team guided us to answer more specific questions. In turn, the data obtained are confidential. Regarding the installation of sensors, we have created a non-invasive technological infrastructure in order to safeguard the privacy of older adults in their homes.

#### 4.5.2. Internal Validity

The threat to the internal validity of our study is the different behavior of each subject in their homes. It is natural that each older adult has his or her routine at home and that there may not necessarily be behavioral patterns that help us infer when an older adult suffers a fall or not. On the other hand, it may be that each older adult understands the instructions differently, i.e., not everyone has the same ability to execute the instructions in the case study. To mitigate these threats, we have studied each patient’s profile and clinical records. We have created a mobile application that allows us to assess the clinical profile of the elderly patient in order to enable them to participate in the study. The results delivered by the mobile application allow us to identify and categorize patients in order to obtain more precise sensor data.

#### 4.5.3. External Validity

The identified threat to external validity is having an unrepresentative group of subjects to generalize the results, i.e., having the wrong subjects to participate in the case study. We have consulted with medical teams on what type of older adults can participate in our research to mitigate this threat. As a result, we identified and characterized variables to select older adults for the case study. These variables were validated by a medical team specialized in older adults.

#### 4.5.4. Conclusion Validity

We have identified heterogeneity of the study group as a potential threat. In case the study group is very heterogeneous, there may be a risk of variation in the results of our study. To mitigate this threat, we have defined inclusion and exclusion criteria to select subjects. These criteria allow us to select homogeneous subjects in order not to affect the validity of our study.

### 4.6. Study Limitations

The geographical area where the sensors and the technical infrastructure of the case study are located leads to several challenges. The cities where the study subjects live are characterized by mountainous terrain. This implies that 3G and 4G signals present connectivity problems in some homes. Therefore, in this study, we selected homes of older adults whose connectivity to 3G and 4G networks were robust enough to conduct the case study. We propose a 5G-based connectivity solution for those homes with problems, which will be addressed in future work.

## 5. Results

### 5.1. RQ1: Classification of Daily Activities of Older Adults

Figure 8 represents the results of the hourly activity score of an older adult in the living room of his house (yellow color) and a bedroom (red color), for each day of August 2020. Through pandas (https://pandas.pydata.org (accessed on 7 March 2022)) and numpy (https://numpy.org (accessed on 7 March 2022)) libraries, we can describe the presence scores through a stakeholder-friendly dashboard. This dashboard is composed of days, weeks, and months.

The calculation of presence scores potentially allows obtaining important information in order to classify the daily activities of older adults. In order to have a better insight about daily activity evolution along the year, we calculate mean hourly activity for each month, i.e., the activity at each hour of the day and each room is averaged over one month. Furthermore, if we extend Figure 8 to a broader perspective, we can identify principles of patterns of daily activities of the older adult. Figure 9 extends the activity scores of the same older adult in Figure 8 for eight months, from January to August 2020, in the same two rooms. This information allows us to observe different phenomena that occur with the activity of the older adult. For example, we can observe that, on average, the hours of most significant activity of the older adult range from 10.00 h and 15.00 h with a peak at 11.00 h. A second example of monthly averaged activity is shown in Figure 10, of another older adult in a different house with less regularity.

Both figures show the difference between an older adult with a lot of movement in a home versus an older adult with little movement. In Figure 9, the charts show higher percentages of activity which fluctuate between 5% to 15%. On the other hand, in Figure 10, the activity percentages range from 1% to 3%. Beyond showing the daily activities of the older adult, the charts represented in Figure 9 and Figure 10 can provide additional information for clinicians. For example, it is striking why the older adult in Figure 10 has a low activity rate. This may be due to the fact that this older adult has other pathologies that do not allow him/her to perform many activities.

### 5.2. RQ2: Fall Detection

Figure 11 and Figure 12 shows examples of results for the same subject who is first standing, then falling, and then on the ground. Figure 11 describes the original infrared sensor image and how the image looks once it is binarized and transformed into a skeleton. On the other hand, Figure 12 illustrates how our approach transforms an image of a senior into a graph-based image. This mapping of the image allows the older adult to be represented from a morphological point of view in order to classify the event as fall/non-fall (see Figure 7).

The validation of the fall detection system was performed using a dataset acquired in our laboratory. This dataset contains 120 falls and 120 non-falls data frames from 10 subjects. The expected precision of the proposed fall detection model is 90.8% (sensitivity: 90.8% and specificity: 90.9%) according to the training performed using the abovementioned dataset. Additionally, a total of 12 falls has been alerted by our falling detection system. From these alerts, the call center confirmed one effective fall (the person had a small fall but did not need assistance). The rest of the alerts corresponded to false positives.

During 2019, out of a total of 12 fall alerts, one effective fall was detected, and 11 false positives were detected as falls events, such as turning on (or moving) a cooker, cleaning, and 8 other events that were not specified in the system by the telephonists. The response time between the detection of the events and the call from the call center (and response from the callers) was, on average, 4 min and 48 s.

During 2020, in order to recognize more movement patterns in falls, the thresholds and the classifier developed were modified, which resulted in several other actions of older adults being detected as false positives, such as coming to rest (sitting or lying down) and picking up objects from the floor. However, during this period, there were no positive alerts as no older adults suffered from falls. In mid-2020 and analyzing the alerts received, the classifier was modified again in order to filter out the actions of older adults that were causing false positives. These modifications were successfully performed, and in the following months, alerts for activities such as picking up objects, sitting, lying down, housekeeping activities, and moving/lighting a cooker were reduced (and even completely filtered out).

## 6. Discussion

The results obtained by the activity scores allow the visualization of periods of time in which the older adult presents an abnormality (possibly a fall) in their daily activities. The data generated by the daily visualization of the older adult’s activities allow us to infer patterns of behavior of the older adult in order to estimate a fall at a certain point in time. In our experience, this information is advantageous in determining protocols and processes for immediate action to assist the older adult. Although the strategy of using activity scores may be straightforward, this method allows us to generate significant knowledge regarding fall prevention. In this regard, some of the results we have analyzed are related to determining trends in the charts illustrated in Figure 8. These charts can give an early indication of when an older adult patient has activity abnormalities in real time. For example, if the charts generated by activity scores calculation decrease in a time interval when there usually is movement on the room, it means that the patient is showing health symptoms that can potentially translate into a fall. Although we cannot generalize that when there is an abnormality in the chart at a specific time interval it is a possible fall of the older adult, we can alert medical staff and the family to this situation in order to make decisions and take preventive actions if required.

On the other hand, Figure 9 and Figure 10 show the hours when the patient is most active monthly. Both charts (blue and black lines) illustrate the older adult’s degree of activity during the day per month. With this information, it is possible to establish the time intervals of highest and lowest activity of the older adult patient (e.g., waking, sleeping, and daily activities at home). At peak activity intervals, the data provided by the charts illustrated in Figure 9 and Figure 10 act as variables for clinicians to study several pathologies in the older adult patient’s speed. The volatility of the charts may be indicative of neurological, circulatory, or musculoskeletal problems in the older adult patient. Earlier detection of these abnormalities allows clinicians to estimate a possible fall in the older adult patient and to determine the risk factors associated with this fall.

Regarding the use of infrared sensors to detect falls through skeletal analysis of the patient, the results obtained in our study reveal that our proposal delivers promising results for fall estimation. The falls detected between 2019 and 2020 allow us to analyze and evaluate the benefits of using infrared sensors as variables to detect and react to falls. In this regard, other studies such as [24] discuss the benefit of using infrared sensors to detect falls. In general, these benefits point to better interpretation of the data, better segmentation of the human body in the sensor image, and non-intrusive, non-contact monitoring. In our study, we corroborate these benefits, but, additionally, our proposal allows us to be more pragmatic and translate these benefits into action protocols in the face of an older adult’s fall.

The skeletal analysis and the diagram of the older adult’s daily activities that we have proposed allow us to define a more precise protocol for informing family members and clinical units in the event of a fall at home. On the one hand, the skeletal analysis gives us the first alert about a human body presenting an anomaly in the patient’s morphology that could be a potential fall. On the other hand, the daily activity analysis reveals, in real time, a low score of the patient in a period of time where the patient usually presents a higher score. These variables are input to define action protocol and alert all stakeholders about a potential fall. This protocol takes data from the daily activity of the older adult and data from the skeletal representation in order to provide early warnings. The protocol considers the following data: (i) the person who receives the fall alert, whether or not they respond to the call, (ii) the start date of the potential fall (the time indicates when the values indicate a potential fall), (iii) the end date (when the older adult was attended to or stood up on their own), (iv) the notification date, and (v) comment on the fall entered by an analyst on the eHomeseniors platform.

The main advantage of our proposal is that we can obtain promising results regarding the detection of falls in older adults through a cost-effective platform based on infrared sensors. The use of this type of sensors allows us to create concrete technological solutions that are able to analyze different behaviors of older adults and, at the same time, optimize costs to be affordable to the budget of older adults. At the same time, the implementation of this platform becomes a support for both the older adult and their social environment, reducing the need for caregivers in their daily activities. On the other hand, the main limitation is related to the geographic difficulty in which the older adults who participated in our study live. The vast majority of the homes are located in a hilly and mountainous area, which limits the communication between the sensors and the eHomeseniors platform. In addition, although older adults accepted the installation of the sensors in their homes, it is not possible to prevent them from disconnecting the sensors themselves, either intentionally or unintentionally. Therefore, an expert clinical accompaniment in the psychological treatment of the older adult is significant to avoid disrupting the communication between the sensors and the eHomeseniors platform and thus not altering the fall detection processes and algorithms.

Regarding the lessons learned in our study, we realized that it is also necessary to know the trajectory of the older adult within the home to improve the results of fall detection. The results obtained in the research questions, together with the entropy in the trajectories of the older adult, can optimize the action plan to detect a potential fall. According to [25,26], if we consider the trajectory of the older adult, we can better detect abnormal behavior patterns that can be associated with possible chronic diseases, such as dementia and nocturia.

To further our research, we aim to use the data from our proposal to calculate the entropy in the trajectories of the older adult to detect abnormal behavior patterns that can be associated with symptoms of chronic diseases. Additionally, we want to evaluate the implementation of gas sensors (e.g., carbon monoxide) for early detection of gas leaks and/or lack of oxygen.

## 7. Conclusions

In this paper, we presented an approach to classify and detect falls of older adults in their homes through infrared sensors. We applied this extension to the eHomeseniors platform, which aims to monitor older adults at home. On the one hand, we proposed an activity measurement algorithm that compares the temperature of each sensor pixel with the temperature to calculate the activity scores of an older adult in a room. On the other hand, we used graph techniques to represent the human body as a skeleton in order to propose an algorithm to establish the fall of an older adult by checking the positions that belong to the skeleton. For both cases, we used non-invasive infrared thermal sensors to respect the older adult’s privacy. We evaluated both methods with a group of older adults who are users of the eHomeseniors platform. The results of our study describe that by combining both methods, it is possible to establish more precise alerts to classify and detect falls. We also identified that the falls detected between 2019 and 2020 were indeed real falls. This implies that the older adult’s families and the emergency network can act early in the event of a fall and thus improve the older adult’s quality of life.

## Figures and Tables

**Figure 1 sensors-22-02321-f001:**
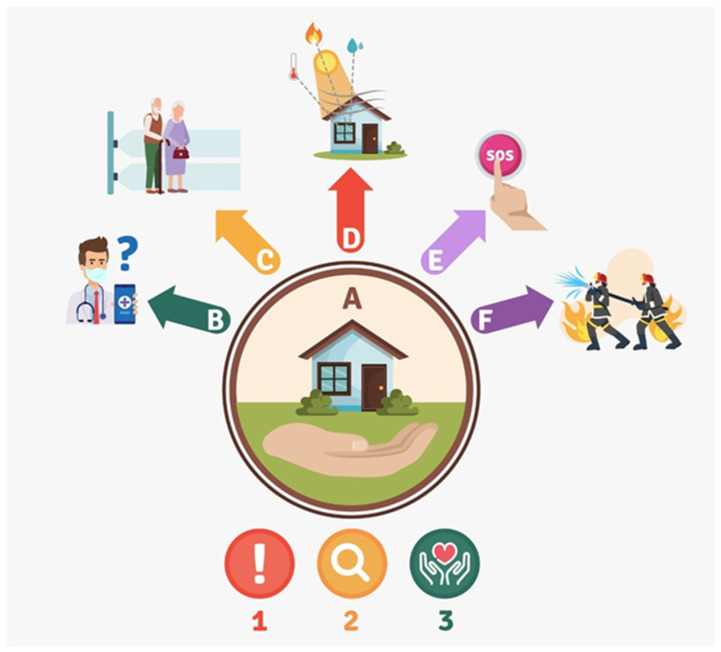
The eHomeseniors platform and its main services provided to older adults.

**Figure 2 sensors-22-02321-f002:**
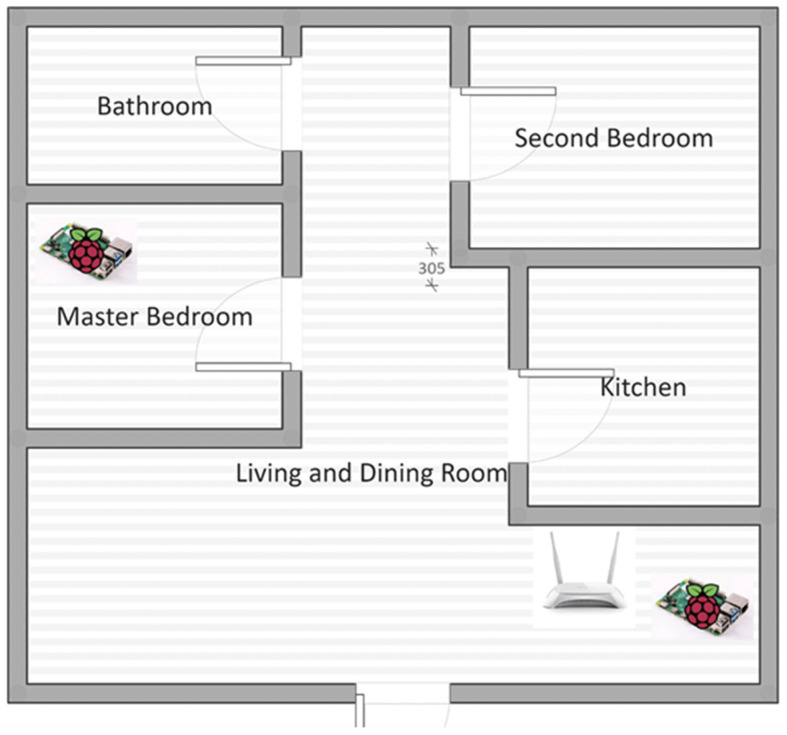
Distribution of sensors in the home of an older adult participating in our study.

**Figure 3 sensors-22-02321-f003:**
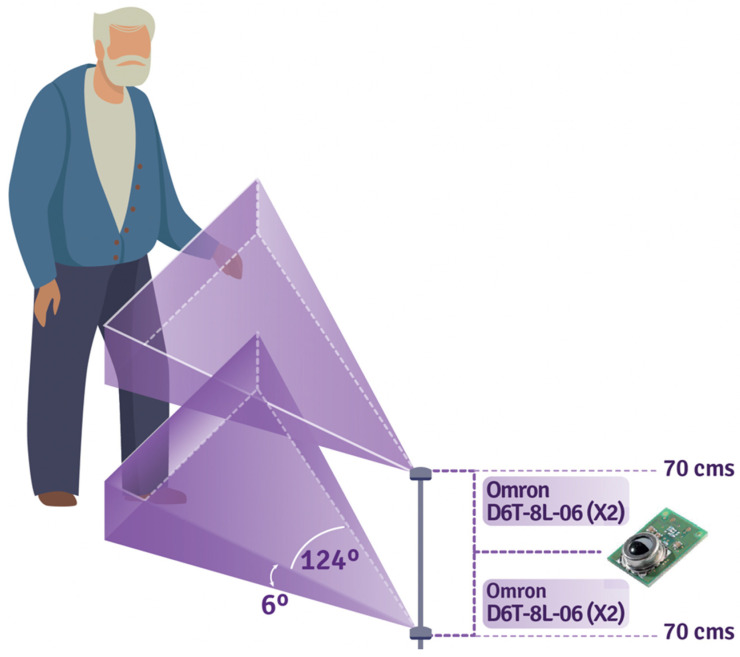
Illustration of the angle of capture of the Omron D6T-8L-06 sensor.

**Figure 4 sensors-22-02321-f004:**
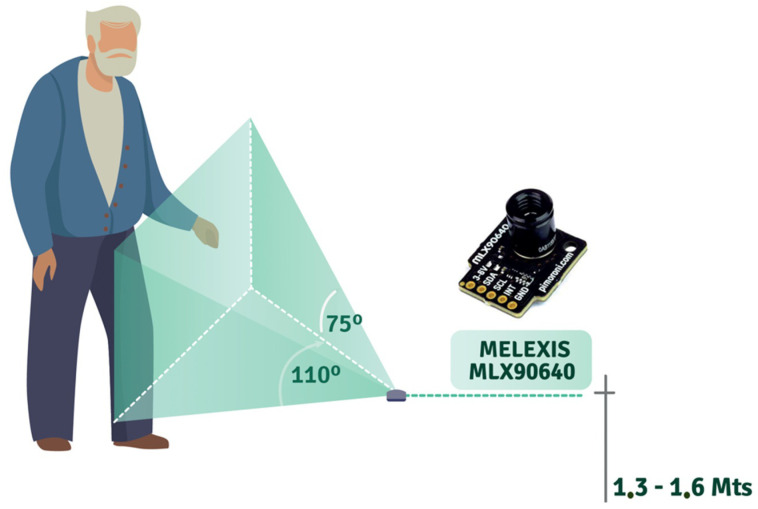
Illustration of the angle of capture of the Melexis MLX90640 sensor.

**Figure 5 sensors-22-02321-f005:**
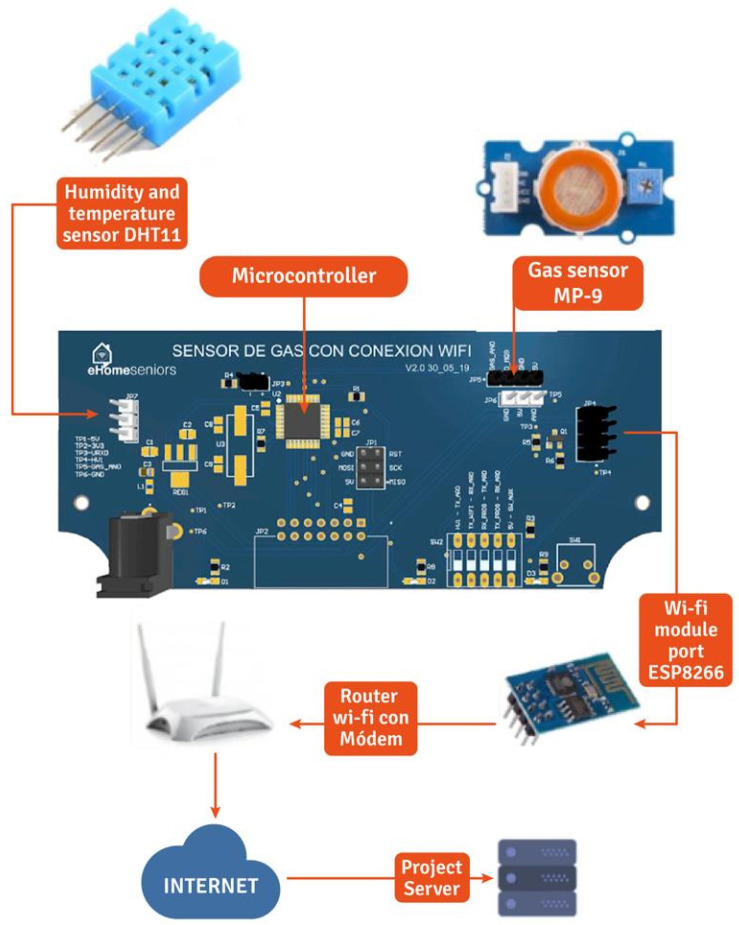
DHT11 sensor and data capture process.

**Figure 6 sensors-22-02321-f006:**
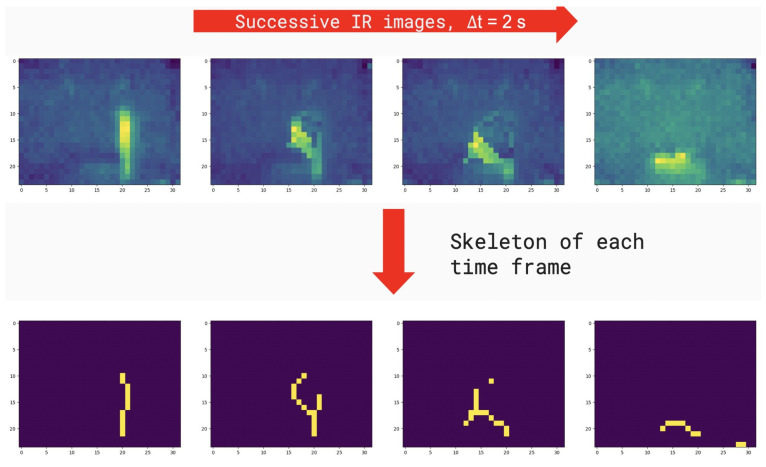
Resulting skeletons of an elderly person that is suffering a fall in front of the infrared sensors. The upper area illustrates how the infrared sensor captures the original image. The lower area describes how the same image looks when the image is skeletonized.

**Figure 7 sensors-22-02321-f007:**
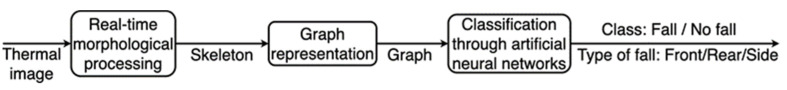
Diagram of the fall detection model.

**Figure 8 sensors-22-02321-f008:**
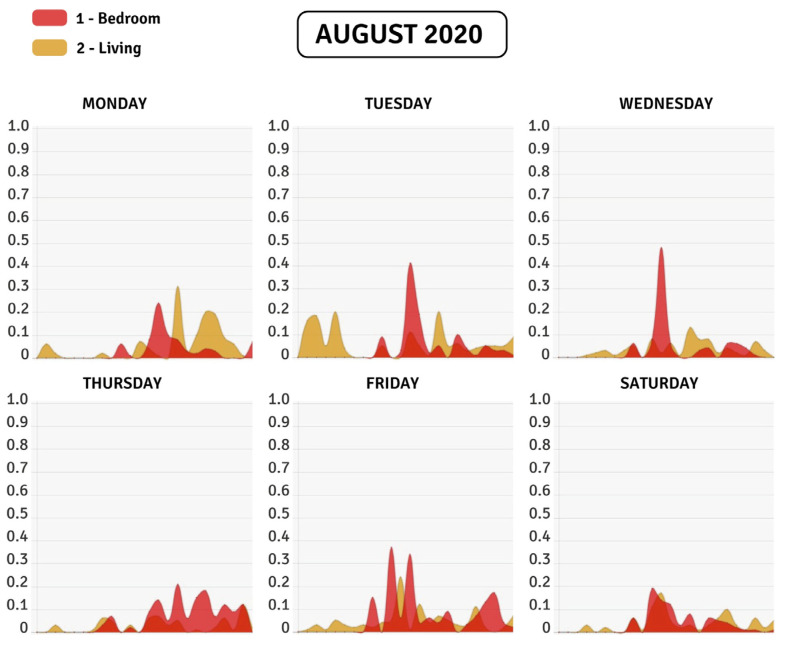
Description of the hourly activity in two rooms within the same house in a week. The red chart represents activity in a bedroom and the yellow chart represents activity in the living room.

**Figure 9 sensors-22-02321-f009:**
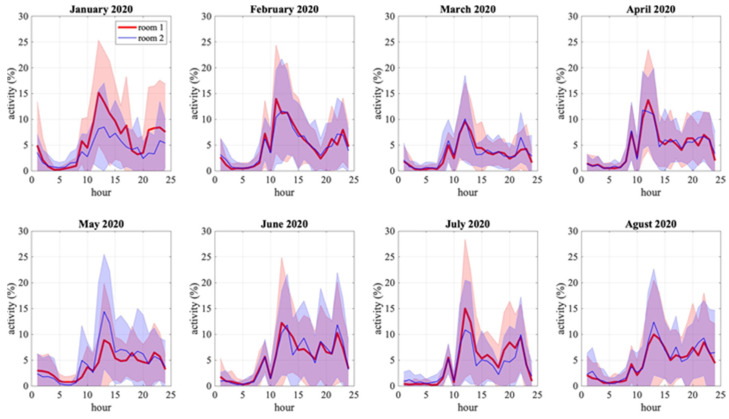
Visualization of the monthly mean activity in two rooms within the same house. The blue and red charts represent different rooms in the home. Error bars, corresponding to the standard deviations, are shown with filled colored areas.

**Figure 10 sensors-22-02321-f010:**
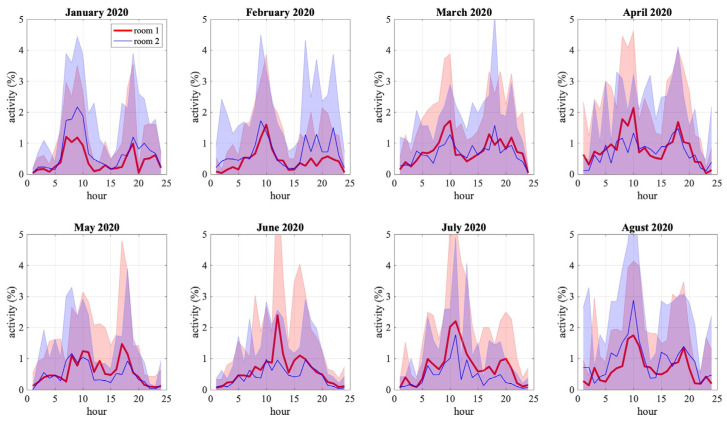
Visualization of the monthly mean activity in two rooms within a second house. The blue and red charts represent different rooms in the home. Error bars, corresponding to the standard deviations, are shown with filled colored areas.

**Figure 11 sensors-22-02321-f011:**
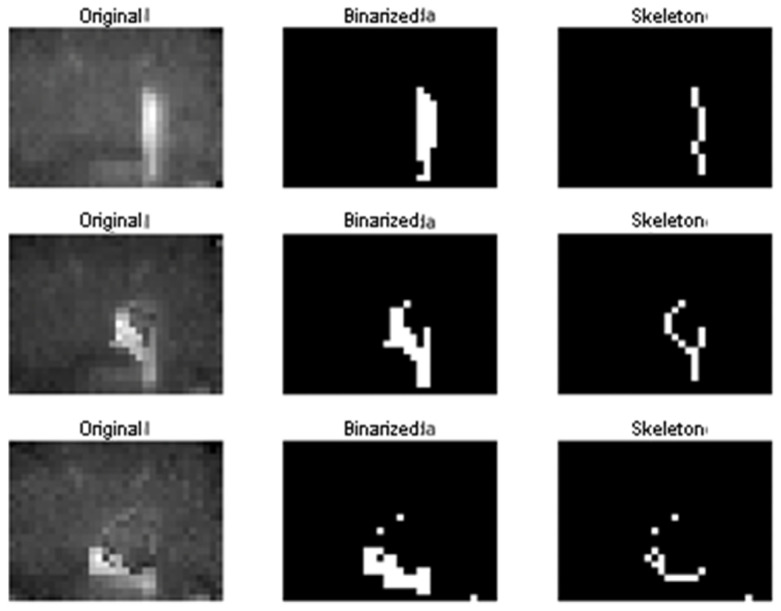
Examples of thresholding and skeleton calculations. The figure represents the mapping of the original image captured by the sensor to its skeletonized representation in three different situations: subject standing, falling, and down.

**Figure 12 sensors-22-02321-f012:**
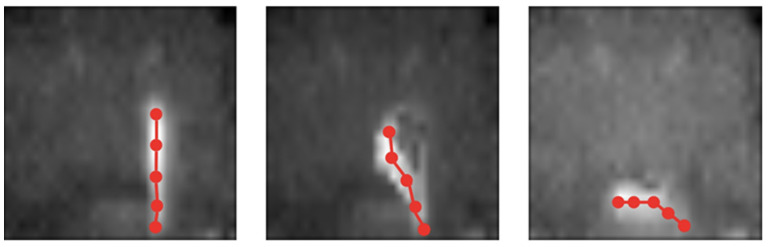
Graph-to-subject adjustment in infrared imaging.

**Table 1 sensors-22-02321-t001:** Sensors high-level description.

Goal	Description
Monitoring of environmental conditions	The platform uses MQ-9 sensors for the detection of gases. It can detect concentrations of gases from 100 ppm to 10,000 ppm. Furthermore, a DHT22 sensor is used to measure humidity and temperature. This sensor has a temperature range from −40 °C to 125 °C with 0.5 °C of precision, and from 0% RH to 100% RH with 5% accuracy for humidity. Moreover, it has a samplingrate of 0.5 Hz (it delivers one measurement every 2 s).
Monitoring accidents	The platform use sensors model AMG8833 of low resolution (8 × 8 pixels in a viewing angle of 60° × 60°) in order to keep the privacy of the user. These sensors are located in each room and are distinguished by an “id”. An ODROID C1+ minicomputer is used to process the information, which is connected to the sensors through an ATMEGA328P microcontroller. This module is adopted to detect possible falls with an accuracy of 90%, using artificial intelligence algorithms.
Monitor of the patient’s health status	Thermal sensors of low resolution used to detect falls are employed for classifying the daily activities performed by the elderly. Then, through a probabilistic model based on Markov chains, it is possible to detect anomalies in the patient’s general behavior pattern.

**Table 2 sensors-22-02321-t002:** Description of the older adults participating in our study.

Age	Gender	Pathologies	Barthel Index
74	Female	Arterial hypertension, diabetes, cardiac arrhythmia	Independent
77	Female	Arterial hypertension, hypothyroidism, chronic obstructive pulmonary disease, arterial hypertension, urinary incontinence, glaucoma	Independent
78	Female	Arterial hypertension, urinary incontinence, glaucoma	Mild dependency
67	Female	Dyslipidemia	Independent
86	Male	Obstructive arteriopathy, coronary heart disease	Mild dependency
78	Male	Arthrosis, arterial hypertension, cardiac arrhythmia due to atrial fibrillation	Independent
76	Female	Diabetes, arterial hypertension, dyslipidemia	Independent
70	Female	Diabetes, arterial hypertension, right hemiparetic disease	Mild dependency
69	Female	Hypertension, insulin resistance	Independent
80	Male	Asthma, high blood pressure, perforated hiatal hernia, dyslipidemia, depression	Mild dependence
83	Female	Cerebral infarction, pernicious anemia	Mild dependency
79	Female	Polio sequelae, diabetes, osteoarthritis	Independent
83	Female	Hypothyroidism	Independent
80	Male	Middle ear problems	Independent
69	Male	Arterial hypertension	Independent
71	Female	Arterial hypertension, arthritis, osteoarthritis, hypothyroidism, glaucoma, hernia, disc disease	Mild dependency

**Table 3 sensors-22-02321-t003:** Description of size, frequency, and format of sensors.

Sensor	Size and Frequency of Capture	Format
Melexis MLX90640	2D, with a frame rate ⇡16 fps	Low resolution image of 32 ⇥ 24 pixels
Omron D6T-8L-06	1D, with a frame rate ⇡ 5 fps	Low resolution image of 1 ⇥ 8 pixels

## Data Availability

Not applicable.

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
