# Peer review of "Using Low-Resolution Non-Invasive Infrared Sensors to Classify Activities and Falls in Older Adults"

_sensors, 2022, doi:10.3390/s22062321_

Round 1
Reviewer 1 Report
This manuscript reports an approach to clasiffy and detect falls of older adults in theri homesusing low-resolution infrared sensors and an eHomeseniors platform. This work could help to detect abnormal behavior of older adults in smart homes. Some points can be considered to improve this manuscript.
1.-Which are main advantages and limitations of the proposed approach in comparison with other methods reported in the literature?
2.-Authors should add more information about main characteristics of the two older adults. For instance, age, weight, height, and so.
3.-More description about geometric distribution of the array of four infrared sensor used for both homes. Was this distribution of sensor the same for both homes? Authors should add images of this distribution of sensors on both homes. Authors must include more detail description of all the stages and experimental setup used the reported approach.
4.-Resolution quality of Figure 4 must be improved.
5.-Quality and size of Figure 5 is poor. The labels of this figure are very small.
6.-Which were the statistical data of the results reported in Figure 6 and 7. For instance, standard desviation.
7.-Authors should add more discussion about difference between the results of figures 6 and 7.
8.-What was the procedure to determine el activity porcentage in Figures 6 and 7? What does represent 15 % on the Figure 6?
9.-More results of the activity of both older adults should be included as suplementary material.
10.-More discussion about reported results on figures 6-9 should be added.
11.-Which are the main challenges of the proposed approach?
12.-More description of the captions of all the figures should be included.
13.-Conclusion must be written in past tense.
Author Response
Dear Reviewer,
Thank you very much for all your comments. We have addressed each of them, which are detailed below.
1.-Which are main advantages and limitations of the proposed approach in comparison with other methods reported in the literature?
Answer: Thank you very much for the comment. We agree with the reviewer with regard to discussing the main advantages and limitations of our proposal compared to the literature. Therefore, we have added the paragraph from line 445 and 467 where we discuss advantages and limitations detected in our study. Additionally, section 4.6 describes the study limitations.
2.-Authors should add more information about main characteristics of the two older adults. For instance, age, weight, height, and so.
Answer: We appreciate the reviewer's comments. After a thorough analysis, we have found that the reviewer is correct in his comment to provide more detailed data on the patients who participated in our study. Therefore, we have added Table 2, describing the age, gender, pathologies and Barthel index of each older adult who participated in our study. We believe that this information is sufficient to clarify the design of our study. Additionally, we have added a new sentence between lines 180 and 186 of the text explaining the table and index used.
3.-More description about geometric distribution of the array of four infrared sensor used for both homes. Was this distribution of sensor the same for both homes? Authors should add images of this distribution of sensors on both homes. Authors must include more detail description of all the stages and experimental setup used the reported approach.
Answer: Thank you very much for your comment. In order to detail and contextualize the reviewer's concern, between lines 187 and 192 we have described the distribution of sensors in households. Additionally, we added Figures 2 to specify further the installation of the sensors in the homes of older adults.
4.-Resolution quality of Figure 4 must be improved.
Answer: Thank you very much for your comment. We have improved the quality of Figure 4 (now, is Figure 7).
5.-Quality and size of Figure 5 is poor. The labels of this figure are very small.
Answer: Thank you very much for your comment. We have improved the quality of Figure 4 (now, are Figure 3 and Figure 4).
6.-Which were the statistical data of the results reported in Figure 6 and 7. For instance, standard desviation.
Answer: Many thanks for your comment. Mean values were initially reported in figures 6 and 7 (now, figures 9 and 10). As the standard deviations are in the same order as the means, error bars have been added with filled colored areas to the figures in order to keep the readability.
7.-Authors should add more discussion about difference between the results of figures 6 and 7.
Answer: Many thanks for the reviewer's comments. There is no doubt that the charts described in the Figures represent part of the results obtained by our study. Therefore, and following the reviewer's suggestion, we have added a paragraph between lines 354 and 361 where we discuss the results of the figures.
8.-What was the procedure to determine el activity porcentage in Figures 6 and 7? What does represent 15 % on the Figure 6?
Answer: Thanks for the comment. The procedure is described in algorithms 1 to 3 (see Appendix A). The activity percentage corresponds to the ratio between measured activity and all measurements for one particular hour.
9.-More results of the activity of both older adults should be included as suplementary material.
Answer: Thank you very much for your comment. To address the reviewer's suggestion, we have added Table 2 to describe the details regarding the older adults who participated in our study.
10.-More discussion about reported results on figures 6-9 should be added.
Answer: Many thanks for your comment. We agree with the reviewer's suggestion. Therefore, we have detailed Figures 6 - 9 (now, Figures 9 - 12) in order to provide the reader with a better understanding of the meaning of the figures.
11.-Which are the main challenges of the proposed approach?
Answer: Thank you very much for your comment. We have analyzed this reviewer's comment and have decided to describe the main challenge we have identified in our study. Therefore, the paragraph between lines 461 and 467 describes the most important challenge we have identified in our research.
12.-More description of the captions of all the figures should be included.
Answer: Thank you very much for your comment. We fully agree with the reviewer. Therefore, we have analysed the captions of each figure in the manuscript and have described in more detail where the description was not sufficient.
13.-Conclusion must be written in past tense.
Answer: Thank you very much for your comment. We have changed the conclusions to past tense.
Reviewer 2 Report
Overall, the paper was well written and provided a compelling case for using IR sensors for activity monitoring and fall detection, although more data is needed from a variety of subjects and living situations. The following are some suggestions for improving the manuscript.
Line 52: change to "allows continuous analysis"
Line 133: change to "through"
Line 144: change to not "interfere"
Line 148: that "they" are being monitored
Page 5: What is the range (distance) of the IR sensors? How many sensors are needed for a room of ____ size? Where are they placed in the room?
Line 200: have "a" score value
figure 2: text in image is unreadable
Line 205: [P]andas
Pages 7,8,9: Not sure actual code should be included in main body of manuscript. Maybe better to put in an Appendix or show diagram of pseudo-code.
Line 234: that is not "falling."
Pages 8, 11: Footnotes are using same numbering as references
Line 344: It appears that only one actual fall was detected over an entire year. Were any falls missed by the system?
Results section: Were only two systems set up, data collected, and analyzed?
Figure 10: Despite translation of what was in text box appearing on page 15, it's not clear what this image says and why it is needed.
Line 207: [N]umpy
Author Response
Dear Reviewer,
Thank you very much for all your comments. We have addressed each of them, which are detailed below.
Overall, the paper was well written and provided a compelling case for using IR sensors for activity monitoring and fall detection, although more data is needed from a variety of subjects and living situations. The following are some suggestions for improving the manuscript.
Answer: Thank you very much for the reviewer's comment. As we have addressed the reviewers' comments, we have added more information and data in order to clarify and detail aspects of the manuscript that remained ambiguous.
Line 52: change to "allows continuous analysis”
Answer: Thank you for your comment. The text has been updated.
Line 133: change to “through"
Answer: Thank you for your comment. The text has been updated.
Line 144: change to not “interfere”
Answer: Thank you for your comment. The text has been updated.
Line 148: that "they" are being monitored
Answer: Thank you for your comment. The text has been updated.
Page 5: What is the range (distance) of the IR sensors? How many sensors are needed for a room of ____ size? Where are they placed in the room?
Answer: Thank you very much for your comment. In order to address the reviewer's comment, we have described the installation of sensors in depth in the paragraph between line 187 and 192. In addition, we have added Figure 2 to illustrate an example of how we organize sensors in houses.
Line 200: have "a" score value
Answer: Thank you for your comment. The text has been updated.
figure 2: text in image is unreadable
Answer: Thank you very much for your comment. We have decided to separate the figure into two (Figure 3 and Figure 4) in order to make the text more readable.
Line 205: [P]andas
Answer: Thank you very much for your comment. After considering the reviewer's suggestion, we have decided to keep the phrase "Pandas" as suggested in the literature.
Pages 7,8,9: Not sure actual code should be included in main body of manuscript. Maybe better to put in an Appendix or show diagram of pseudo-code.
Answer: Thank you for your comment. We have followed the reviewer's suggestion to re-organize the pages mentioned by the reviewer. Therefore, we have added an appendix detailing the algorithms mentioned on pages 7, 8 and 9.
Line 234: that is not “falling."
Answer: Thank you for your comment. The text has been updated.
Pages 8, 11: Footnotes are using same numbering as references
Answer: Thank you for your comment. The text and numbering have been updated.
Line 344: It appears that only one actual fall was detected over an entire year. Were any falls missed by the system?
Answer: Thank you very much for your comment. We have described between lines 370 and 377 more detail regarding the data reviewed during one year. We also detail quantitative data on the total number of falls detected and lost by our proposal.
Results section: Were only two systems set up, data collected, and analyzed?
Answer: Thank you very much for your comment. Our platform is made up of other sensors that help us collect more data to have the most accurate results possible. In order to further detail our sensor infrastructure, we have described between paragraphs 238 and 244 the DHT11 sensor that captures additional data for our platform. Additionally, we have added Figure 5 which describes the sensor and the data capture process.
Figure 10: Despite translation of what was in text box appearing on page 15, it's not clear what this image says and why it is needed.
Answer: Thank you very much for the reviewer's comment. After analyzing the figure, we agree with the reviewer that Figure 10 does not add to the discussion of the paper. Therefore, we have decided to remove the figure from the manuscript and re-organise the paragraph from lines 431 and 444.
Line 207: [N]umpy
Answer: Thank you very much for your comment. After considering the reviewer's suggestion, we have decided to keep the phrase "Pandas" as suggested in the literature.
Round 2
Reviewer 1 Report
Authors have improved their manuscript considering all the comments of reviewer.